# Multistage BiCross encoder for multilingual access to COVID-19 health information

**Iknoor Singh** *, **Carolina Scarton**, **Kalina Bontcheva**

Department of Computer Science, University of Sheffield, Sheffield, United Kingdom

* i.singh@sheffield.ac.uk

## Abstract

The Coronavirus (COVID-19) pandemic has led to a rapidly growing 'infodemic' of health information online. This has motivated the need for accurate semantic search and retrieval of reliable COVID-19 information across millions of documents, in multiple languages. To address this challenge, this paper proposes a novel high precision and high recall neural Multistage BiCross encoder approach. It is a sequential three-stage ranking pipeline which uses the Okapi BM25 retrieval algorithm and transformer-based bi-encoder and cross-encoder to effectively rank the documents with respect to the given query. We present experimental results from our participation in the Multilingual Information Access (MLIA) shared task on COVID-19 multilingual semantic search. The independently evaluated MLIA results validate our approach and demonstrate that it outperforms other state-of-the-art approaches according to nearly all evaluation metrics in cases of both monolingual and bilingual runs.

**Data Availability Statement:** The data underlying the results presented in the study are available at http://eval.covid19-mlia.eu/task2/.

## 1 Introduction

The COVID-19 pandemic has, to date, infected more than 135M people worldwide. It has also been accompanied by what the World Health Organisation has dubbed an 'infodemic', in reference to the challenge people face in navigating and absorbing the continuously growing volumes of information on the origin, treatment, prevention, and public policies related to COVID-19 that get published online by numerous sources (some authoritative and some not), in multiple languages and countries. This has prompted the need for new efficient and accurate multilingual semantic search and retrieval methods for health information. To facilitate the comparative evaluation of existing approaches and research on new methods, the COVID-19 Multilingual Information Access (MLIA) shared task [1] released benchmark datasets of COVID-19 related information spanning all EU languages and beyond. These datasets address three key information access tasks: Information Extraction (Task 1), Multilingual Semantic Search (Task 2), and Machine Translation (Task 3). Here we focus specifically on the multilingual semantic search task, which provides a large multilingual dataset to support research and a comparative evaluation of semantic search approaches (with the purpose of analysing and improving the retrieval of relevant COVID-19 documents from a given user query).

**Funding:** This work has been partially supported by the WeVerify and SoBigData++ projects 485 (EU H2020, grant agreements: 825297 and 871042).

**Competing interests:** The authors have declared that no competing interests exist.

In the past few years, the large pre-trained transformer models such as BERT [2], RoBERTa [3], XLM [3, 4] and others have achieved state-of-the-art performance on a wide range of natural language processing tasks from semantic text similarity to question answering. Recently, these have also been applied to information retrieval tasks [5–8]. In this paper, we present a novel multistage BiCross encoder method and demonstrate that it outperforms other state-of-the-art retrieval methods for COVID-19 multilingual semantic search, according to independent comparative evaluation on the MLIA shared task 2 dataset.

Multistage BiCross encoder is a sequential three-stage ranking pipeline, composed of: 1) a BM25 retrieval stage 2) a neural refinement stage, and 3) a neural re-ranking stage. Our approach exploits both bi-encoder and cross-encoder transformer architectures to compute the document-level relevance score by aggregating sentence-level relevance scores. For document retrieval, cross-encoders tend to attain significantly higher accuracy [8], due to the rich interactions and self-attention over the query and document pair. On the other hand, when the number of documents to be re-ranked is large, cross-encoder recomputes encoding each time during inference [9] which makes them very resource-intensive when compared to bi-encoder which can make use of cached representations for faster inference. Since the performance gains come at a steep computational cost, we use both bi-encoder and cross-encoder with the former having more documents to re-rank as compared to the latter. This way we are able to utilise the benefits of both bi-encoder and cross-encoder neural models on top of the BM25 lexical model. To the best of our knowledge, ours is the first paper to investigate this for document retrieval. The main research question addressed in this paper is: how to improve the architecture and performance of state-of-the-art neural models for document retrieval to make them better suited to retrieving COVID-19 health information in multiple languages?

The key contributions of this paper are:

- Multistage BiCross encoder method, which is a three-stage ranking pipeline that uses the Okapi BM25 retrieval algorithm and state-of-the-art multilingual transformer-based bi-encoder and cross-encoder by aggregating sentence-level relevance scores for the task of COVID-19 multilingual semantic search.

- We experiment with different types of search queries in order to establish the best performing ones for retrieving COVID-19 health information across millions of documents, in multiple languages. We also present ways to combine scores from different stages using various rank fusion algorithms.

- An extensive comparison of our runs with other participant runs to demonstrate the effectiveness of our methods in achieving high precision for top ranked documents, as well as high recall for all retrieved documents in both monolingual and cross-lingual search settings.

- In order to make our findings available for reproduction and to benefit researchers, we publish the trained models and source code at: https://github.com/iknoorjobs/Multistage-BiCross-Encoder.

Prior to introducing the proposed approach (Section 4), we first introduce the multilingual COVID-19 semantic search task and the accompanying dataset provided by the organisers of the MLIA shared task evaluation challenge (Section2). Next, Section 3 discusses previous work on neural methods for semantic search. Section 4 gives a detailed description of the underlying architecture of our Multistage BiCross Encoder and our training methodology. In addition, it also describes the use of external training datasets which improved the model's performance further and helped it achieve the best reported scores on the MLIA COVID-19 semantic search

task, according to the independent comparative evaluation reports by the shared task organisers https://bitbucket.org/covid19-mlia/organizers-task2/src/master/ [10]. Section 5 provides details of all our experimental runs and settings, as well as information on the other participating systems in MLIA. The evaluation results are presented in Section 6, followed by a conclusion in Section 7.

## 2 MLIA COVID-19 semantic search task

Multilingual Information Access (MLIA) COVID-19 is a shared evaluation task run by an independent consortium of researchers and is endorsed by the European Commission and the European Language Resource Coordination (ELRC). In this section, we describe the COVID-19 MLIA dataset and introduce the specifics of the multilingual semantic search task 2. The core challenge of this task is to improve information exchange about COVID-19 in both monolingual and cross-lingual search settings.

### 2.1 Dataset

The dataset consists of a corpus of 3,750,588 documents and a set of 30 query topics, both available in multiple languages: English, French, German, Greek, Italian, Spanish, Swedish and Ukrainian. These languages are spoken in countries where there was a rapid spread of COVID-19 or the pandemic was managed differently at the beginning of 2020. For each language, the corpus consists of general health-related articles collected from different websites, out of which the majority of articles come from the Medical Information System (MEDISYS) https://medisys.newsbrief.eu/medisys/clusteredition/en/24hrs.html. Table 1 shows the number of documents in each language.

There are a total of 30 query topics that are used for querying the dataset. Each query topic comprises of three fields: (i) a keyword field: a set of relevant keywords related to the query; (ii) a conversational field: the query in the form of a question; and (iii) an explanation field: a more detailed description of information that is needed in the retrieved documents. For our experiments, we only used the keyword and conversational fields as the explanation field is more useful for assessing the relevance of the document at evaluation time. Table 2 shows the keyword, conversational, and explanation fields for one of the English query topics on the use of ultraviolet light to kill coronavirus.

### 2.2 Task description

In the MLIA multilingual semantic search (task 2) [10], participating systems need to search the growing information related to the novel coronavirus, in different languages and with

**Table 1. Count of documents for each language in the MLIA corpus.**

| Language | Count |
|---|---|
| English (en) | 1,452,240 |
| Spanish (es) | 833,763 |
| Italian (it) | 662,789 |
| French (fr) | 326,599 |
| German (de) | 273,761 |
| Greek (el) | 147,658 |
| Swedish (sv) | 38,196 |
| Ukrainian (uk) | 15,582 |
| Total | 3,750,588 |

**Table 2. The keyword, conversational, and explanation fields for the MLIA query topic about the use of ultraviolet light to kill coronavirus.**

| Topic Field | Text |
| --- | --- |
| Keyword | uv light to kill coronavirus |
| Conversational | Is uv light effective to kill coronavirus? |
| Explanation | Seeking studies that discuss whether ultraviolet light is an effective way to sanitise against COVID-19 |

different levels of knowledge about a specific topic. The task follows a CLEF-style [11] evaluation methodology where participants are provided with a collection of documents and a set of topics that are used as queries to produce various runs that could either be monolingual or bilingual, depending on the language of the query and the retrieved documents. There are two subtasks: subtask 1 is focused on high precision whilst subtask 2 is oriented towards high-recall systems. Each participating team could submit a maximum of five monolingual and five bilingual runs for each language for each subtask. In monolingual runs, the language of both query and documents is the same whereas for bilingual runs, the language of both query and documents is different.

In order to carry out a comparative evaluation of the participating systems on unseen data, the organisers created an additional pool of around 6000 to 8000 documents for each language by selecting the top $k$ documents from all the runs. These pools of documents were then manually annotated by the experts to produce relevance judgements. The organisers then evaluated all runs using established information retrieval metrics, including recall, precision (P@5 & P@10), R-precision (RPrec), average precision (AP), and normalised discounted cumulative gain (NDCG).

## 3 Related work

Neural models for information retrieval are generally used in a two-stage pipeline architecture where re-ranking is done only on the top results retrieved by traditional ranking methods such TF-IDF or BM25 [5, 6, 8], as the computational cost of running neural models over the entire dataset can be prohibitively high [12]. BM25 [13] is a bag-of-words retrieval model that retrieves documents based on lexical overlap with the query terms. Nogueira et al. [8] are the first to demonstrate that the BERT model can also be used for fine-tuning passage re-ranking tasks and it has shown to be effective for ad-hoc document ranking. They use a sequence of tokens by concatenating the query tokens and the passage tokens, separated by a [SEP] token as an input to the BERT model and then the output embedding of the [CLS] token is passed to a single layer neural network to obtain the probability of the passage being relevant to the query. On the other hand, Karpukhin et al. [6] use dual-encoder (or bi-encoder) architecture to apply FAISS [14] on the encoded BERT representations of questions and passages. Although FAISS makes it fast, due to separate representations, it lacks attention between the question and passage tokens which makes it less accurate [8].

In addition, researchers tried various methods to improve the effectiveness of neural models in document re-ranking tasks. Nogueira1 et al. [15] use a query generator model to expand the document before indexing to get additional gains on the retrieval performance. Other work uses rank fusion methods [16, 17] to combine various runs in order to improve the performance of retrieval systems. Clipa and Di Nunzio [18] analyse and compare various state-of-the-art information retrieval methods and ranking fusion approaches for the domain of medical publication retrieval. Pradeep et al. [19] formulate it as a pointwise and pairwise

classification problem using the sequence-to-sequence T5 model [20] to show its effectiveness in neural re-ranking. Yilmaz et al. [5] use sentence-level evidence to compute document-level relevance scores using a cross-encoder model for ad hoc retrieval. Previous study [21] also shows that the highest scoring sentence in a document is a good indicator of the relevance of the document and it helps in achieving high recall. But applying sentence-level inference on all the documents is also not possible given the computational overhead of transformer models. As far as sentence-pair scoring tasks are concerned, due to transformer attention mechanism [9, 22], cross-encoder are more accurate than bi-encoder. On the other hand, cross-encoders recompute encoding each time during inference as compared to bi-encoder which can make use of cached representations for faster inference. Hence, instead of applying a cross-encoder model on each sentence during inference, we use a bi-encoder to compute the document-level relevance score by aggregating sentence-level relevance scores using cached sentence representations. This is done in the neural refinement stage (Section 4.2) which succeeds BM25 retrieval stage. In the last stage (Section 4.3), a cross-encoder architecture is used which exploits the self-attention of the transformer model to re-rank a subset of candidate documents so as to make relevant documents rank higher. This way we can take advantage of both bi-encoder and cross-encoder for finding semantically relevant documents from the initially retrieved documents BM25 lexical model.

Furthermore, to produce good sentence representations from transformer models, Reimers et al. [23] propose SBERT, where they train BERT-based models using siamese network architecture to get semantically meaningful sentence representations. These can be leveraged for other tasks such as semantic search where the sentence representations can be compared using sentence-pair scoring function. We used this training methodology to train the bi-encoder in the neural refinement stage to bring both query and the relevant document into the proximity of each other in the high dimensional vector space (Section 4.2).

The outbreak of COVID-19 has led to ever-expanding research in deep learning for COVID-19 [24] and healthcare [25]. Recently, the TREC-COVID challenge [26] invited participants to develop information retrieval systems for scientific literature containing tens of thousands of scholarly articles related to COVID-19. Although, both the TREC-COVID challenge and MLIA task 2 involve the development of information retrieval systems for COVID-19 information, the domain of the corpora in both tasks is different: the former is targeted towards scientific scholarly papers whilst the latter is targeted towards systems that provide general health-related articles of relevance to citizens and public health. In addition, the TREC-COVID challenge only had English query and documents whereas MLIA is a multilingual task in which both query and documents are in multiple languages as discussed in Section 2.

## 4 Multistage BiCross encoder

Multistage BiCross encoder is a three-stage ranking pipeline that includes an initial lexical retrieval stage followed by two neural-based semantic retrieval stages. Fig 1 illustrates the architecture of our approach where $q$ is the query, $s_i$ is the $ith$ sentence of the document, model $M1$ is used as a bi-encoder in neural refinement stage (Section 4.2) and model $M2$ is used as cross-encoder in the neural re-ranking stage (Section 4.3).

For lexical retrieval, Okapi BM25 [13] is used to reduce the search space from a large number of documents (e.g. 1.4M in the case of English documents) to a small set of possibly relevant documents. In the second stage (referred to as the neural refinement stage), we leverage a transformer-based bi-encoder model to encode both query and document individually into deep contextualised representations and use them to efficiently re-rank the retrieved

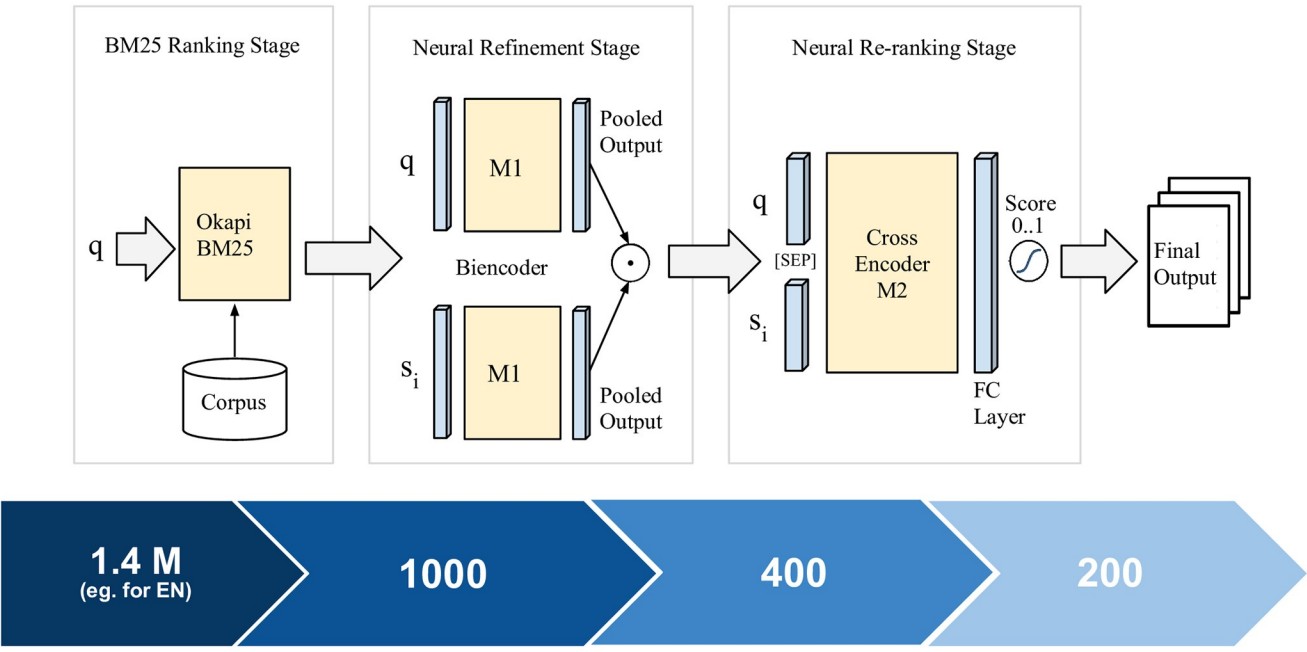

**Fig 1. Overview of Multistage BiCross encoder.** The blue bar below shows top $k$ candidate documents ranked in each stage.

documents based on their relevance. The final neural re-ranking stage uses a transformer-based cross-encoder [8] to re-rank a subset of top-ranked candidate documents output from the neural refinement stage. We also explore various rank fusion techniques to combine the output from the different stages to get a single relevance score which is used for sorting the final list of documents. The blue bar in Fig 1 shows top $k$ candidate documents ranked in each stage. The detailed description of all three stages can be found in the subsequent subsections.

## 4.1 BM25 retrieval stage

The initial set of candidate documents are retrieved using the traditional Okapi BM25 [13] lexical retrieval model as shown in Fig 1. First, we pre-process all documents in the corpus and index them using Elasticsearch https://www.elastic.co/elasticsearch/. Documents belonging to different languages are indexed separately. In our case, we only considered English, Spanish, French and German documents since working on all languages was not feasible within the very constrained timeline set by the organisers for submitting runs.

The corpus provided by the organisers contains documents in the XML format and we have only used the text inside the <p> tags (all boilerplate tags are removed). Text pre-processing methods such as stopword removal and lemmatisation were applied before indexing the documents. As described in Section 2, each query topic has three different fields expressing the information needed in various levels of detail. Our experiments use a concatenation of the keyword and conversational fields (*key_conv*) as a query to retrieve matching documents from the Elasticsearch indexes using BM25.

We also used the keyword and conversational field (*key_conv*) to generate three more queries using sequence-to-sequence T5-base doc2query model [15] and we concatenate these three generated queries with the *key_conv* to form a single query, called hereafter *t5_query*. The idea behind the *t5_query* is to assess the performance of concatenated reworded queries on the MLIA corpus. For example, *t5_query* for the topic query mentioned in Table 2 is "uv

light to kill coronavirus Is uv light effective to kill coronavirus? Does uv light kill coronavirus? Can uv lights kill coronavirus? Is uv light effective against coronavirus?". In addition, we also tried the Udels query from TREC-COVID [27] to evaluate its effectiveness on the MLIA dataset. The Udels query [27] is made up of non-stopwords from the keyword field and the named entities mentioned in the conversational field. For instance, Udels query for the topic query mentioned in Table 2 is "uv light kill coronavirus effective kill coronavirus".

The BM25 ranking stage will filter out all lexically dissimilar documents concerning the query. The score of a document $d$ using Okapi BM25 algorithm is formulated as

$$\text{BM25Score}(d) = \sum_{i=1}^{n} \text{IDF}(q_i) \cdot \frac{f(q_i, d) \cdot (k_1 + 1)}{f(q_i, d) + k_1 \cdot \left(1 - b + b \cdot \frac{|d|}{\text{avgdl}}\right)} \tag{1}$$

where $BM25Score(d)$ is the BM25 relevance score of document $d$, $q_i$ are token keywords of the given query, $IDF(q_i)$ is the inverse document frequency of the query term $q_i$, $f(q_i, d)$ is the frequency of query term in the document, $|d|$ is the number of words in the document $d$, $avgdl$ is the average length of documents in the complete database and the rest are the default parameters ($k1 = 1.2$ and $b = 0.75$) as set in Elasticsearch.

## 4.2 Neural refinement stage

In the second stage, the top 1000 documents retrieved by BM25 are re-ranked using a bi-encoder which is based on Siamese networks [23]. We use a pre-trained transformer-based model to encode both document and query separately into fixed-length deep contextualised embeddings by using mean pooling on the output layer. In the same vector space, query and relevant document lie in proximity to each other and can be efficiently retrieved using cosine similarity as shown in the neural refinement stage of Fig 1. As the representations are separate, the encoded representations from query and document are cached so that they can be reused for faster predictions during inference time. Following [5, 28], each document in the corpus is split into sentences, and we apply inference on each sentence separately to obtain a sentence-level relevance score for each pair of input query and sentence. As the documents in the MLIA corpus are long, we only consider the first N sentences for inference, where N denotes average number of sentences in documents of the corpus. Moreover, previous research [29, 30] shows that any relevant document is likely to contain relevant sentences at the beginning of the document. The document-level relevance score is determined by aggregating the top $k$ scoring sentences in the document as follows:

$$BiScore(d) = \sum_{i=1}^{k} w_i \cdot S_{Biencoder_i} \tag{2}$$

where BiScore(d) is the document-level relevance score for document $d$ using the bi-encoder model. $k = 3$ as shown in [28] and $S_{Biencoder_i}$ is the $i$-th top sentence-level relevance score with respect to the query. Similar to [28], the parameters $w_i$ are tuned via exhaustive grid search. Due to lack of relevance labels for the MLIA task, we have set the initial parameters such that $w_1 > w_2 > w_3$, because we want to give more weight to the sentence which is more relevant as compared to the less relevant sentences. In other words, high scoring sentences contribute more to the final relevance score of the document.

Since there are no relevance labels available for the MLIA task, we generate pseudo-qrels using external datasets specific to the COVID-19 domain which we used to train our models. We prepared the TC+IFCN data which is a combined version of the TREC-COVID challenge

(TC) [26] dataset and the IFCN dataset [31]. The TREC-COVID dataset has 69,318 relevance assessments for 50 different topics. Although the corpus used in the TREC-COVID challenge consists of scientific scholarly articles, we use this dataset to transfer knowledge to our model and test its performance on the MLIA corpus which comprises general health-related articles. On the other hand, the IFCN dataset consists of around 7000 COVID-19 misinformation claims debunked by members of the International Fact-Checking Network (IFCN). In this case, we consider each claim as the pseudo-query and its corresponding fact-checked article body as its relevant document. As we apply inference on each sentence separately, we prepare a sentence-level dataset in which, for each query, we extracted sentences from the document that share something meaningful with the query. For this, we use state-of-the-art models trained on the Semantic Textual Similarity (STS) [32] data to generate both positive and negative sample sentences from the document and we assume that these are relevant with respect to the query. This ensures that training is carried out on the optimal information signal to bring the query and the relevant sentence in the document together. Finally, we also develop a cross-lingual dataset (Cross_TC+IFCN), where we augmented the TC+IFCN dataset by translating the query and document pairs to Spanish, French, and German using OPUS-MT [33]. Cross_TC+IFCN is used to fine-tune multilingual models which are employed for runs that involve documents in languages other than English.

For training the bi-encoder, we utilised the models provided by the sentence-transformers https://github.com/UKPLab/sentence-transformers library, which includes BERT-based models [23] fine-tuned using siamese and triplet networks to get semantically meaningful sentence representations. We further fine-tuned the SBERT models on our domain-specific dataset. The details of the models used in our experiments are as follows:

- For monolingual English runs, we try two models: 1) msmarco-distilroberta-base-v2, RoBERTa [3] base model trained on MSMARCO passage ranking dataset and 2) stsb-roberta-large, a RoBERTa large model trained on natural language inference and semantic textual similarity dataset. We use these as base models to fine-tune on the TC+IFCN dataset with a regression objective function [23]. The final models are referred to as TCIN-msmarco-distilroberta-base and TCIN-stsb-roberta-large.

- For bilingual runs, we use paraphrase-xlm-r-multilingual-v1 [23] which is a xlm-roberta-base [3] model trained on a large scale paraphrase dataset of more than 50 languages. Transfer learning is used to fine-tune this model on Cross_TC+IFCN dataset using the same objective function as mentioned above and the final multilingual model is named as Cross-TCIN-xlm-r-paraphrase.

We call this the neural refinement stage since it helps to filter out all semantically unrelated documents and it also works much faster when compared to a cross-encoder-based approach where a pair of sentences are passed together to the model every time during inference.

## 4.3 Neural re-ranking stage

In the third stage, the top 400 documents retrieved by the neural refinement stage are re-ranked using a cross-encoder architecture. In this, both query tokens and the document tokens separated by [SEP] token are passed to the transformer-based model to perform full self-attention over the given input and the output of [CLS] token is passed to the linear layer with sigmoid activation to get a relevance scores from 0 to 1 [8] as illustrated in the neural re-ranking stage of Fig 1. Similar to the neural refinement stage, here also we split the document into sentences and apply sentence-level inference using the query. The final relevance score for each

document is determined by combining the top $k$ scoring sentences of the document i.e.

$$CrossScore(d) = \sum_{i=1}^{k} w_i \cdot S_{Cross_i} \tag{3}$$

where $CrossScore(d)$ is the document-level relevance score for document $d$ using the cross-encoder model, $S_{Cross_i}$ is the $i$-th top sentence score and all other parameters are kept the same as in Eq 2. Although cross-encoder is more accurate than bi-encoder, they are compute-intensive and time-consuming when compared to bi-encoder and this is the reason we give fewer documents to re-rank in the neural re-ranking stage. The description of models used as cross-encoders is as follows

- For monolingual English runs, we use ELECTRA model fine-tuned on the MSMARCO dataset from [34] as it was among the top positions in the second round of TREC-COVID task. The model was further fine-tuned on the TC+IFCN data using binary cross-entropy loss. We call this model as TCIN-electra-msmarco.

- For bilingual runs, currently, there does not exist any multilingual model trained on the MSMARCO passage ranking dataset. Hence, we use state-of-the-art multilingual transformer-based models such as xlm-roberta-base [3] and distilbert-base-multilingual [2] as a base model for fine-tuning on the MSMARCO passage dataset for an epoch with a learning rate of 1e-5, batch size of 16, and a maximum of 512 input sequence length. The models are further fine-tuned on Cross_TC+IFCN dataset using sigmoid cross-entropy loss and the final models are referred to as CrossTCIN-xlm-roberta-msmarco and CrossTCIN-distilbert-multilingual-msmarco.

In the case of bilingual runs, where query and documents are in a different language, we simply use Google Translate to translate the query into the target language of the document and apply the same methods as described above in a monolingual setting. For some runs, we also combine scores from different stages using various rank fusion algorithms. These can be broadly classified into score-based and rank-based fusion algorithms. For the score-based method, we use weighted CombSUM which is a slight modification of CombSUM [16] algorithm where we add the weighted scores from different ranking models. The equation is as follows

$$\begin{aligned} wCombSUM(d) = \alpha \cdot norm(CrossScore(d)) + \beta \cdot norm(BiScore(d)) \\ + (1 - \alpha - \beta) \cdot norm(BM25Score(d)) \end{aligned} \tag{4}$$

where $wCombSUM(d)$ is the weighted CombSUM score of the document $d$, $norm$ is the min-max normalisation of relevance scores. $BM25Score(d)$, $BiScore(d)$ and $CrossScore(d)$ are the relevance scores of document $d$ from first, second and third stage respectively. The parameters $\alpha$ and $\beta$ are such that $\alpha > \beta$ and we have fixed $\alpha = 0.5$ and $\beta = 0.4$, giving more weight to cross-encoder ranking followed by bi-encoder and BM25 ranking respectively. For rank-based fusion methods, we tried Reciprocal Rank Fusion (RRF) [17] and Borda Fusion [35]. In this, the fused score of the document simply relies on the rank of the document from different ranking models, hence in our work we only use the output of neural refinement and neural re-ranking stage. The equation of RRF and Borda fusion is as follows

$$RRFScore(d) = \frac{1}{k + R_{Cross}(d)} + \frac{1}{k + R_{Biencoder}(d)} \tag{5}$$

$$BordaScore(d) = \frac{N - R_{Cross}(d) + 1}{N} + \frac{N - R_{Biencoder}(d) + 1}{N} \tag{6}$$

where *RRFScore*(*d*) and *BordaScore*(*d*) are the RRF and Borda fusion scores. $R_{Biencoder}(d)$ and $R_{Cross}(d)$ are the ranks of the document *d* from neural refinement and neural re-ranking stage. For the RRF method, we set the constant *k* = 60 default as mentioned in their respective paper [17]. In Borda fusion, *N* is the total number of documents during fusion.

## 5 Implementation details

We implement BiCross encoder to check its effectiveness in both monolingual and bilingual search settings. Our system is designed such that it aims at achieving both high precision as well as high recall. We submitted a total of 22 monolingual runs and 15 bilingual runs. The runs differ in terms of models, type of query, and rank fusion methods. The description of all the runs is given below and all runs retrieve 200 documents for each query. All our runs have `gatenlp_` as a prefix in the name of the run. The experiments were conducted on NVIDIA Titan RTX GPU.

- `gatenlp_run1` / `gatenlp_run2` / `gatenlp_run3`: In run 1, Udels method is used to generate the query, *t5_query* is used in run 2 and a concatenation of the keyword and conversational (*key_conv*) field in run 3. The bi-encoder is TCIN-msmarco-distilroberta-base and cross-encoder is TCIN-electra-msmarco. The encoder models are kept the same in all the runs to see which type of query gives the best results.

- `gatenlp_run5` / `gatenlp_run7`: In run 5 and run 7, we use a different bi-encoder i.e. TCIN-stsb-roberta-large and the cross-encoder model is TCIN-electra-msmarco. The only difference between both the runs is that run 5 uses *key_conv* query and run 7 uses Udels query.

- `gatenlp_es_run25` / `gatenlp_fr_run26` / `gatenlp_de_run27` / `gate-nlp_es_run28` / `gatenlp_fr_run29` / `gatenlp_de_run30`: These are monolingual Spanish, French and German runs. Here, we use multilingual models where bi-encoder is CrossTCIN-xlm-r-paraphrase and cross-encoder is CrossTCIN-distilbert-multilingual-msmarco. In case of run 25, 26 and 27, weighted CombSum fusion is used whereas for run 28, 29 and 30, RRF fusion is used to get the relevance score for each document. In all these runs and the following monolingual runs, *key_conv* query is used to retrieve the documents. The evaluation results of these runs will depict the best performing rank fusion algorithm.

- `gatenlp_es_run31` / `gatenlp_fr_run32` / `gatenlp_de_run33` / `gate-nlp_es_run34` / `gatenlp_fr_run35` / `gatenlp_de_run36` / `gatenlp_es_-run37` / `gatenlp_fr_run38` / `gatenlp_de_run39`: In the above runs, we use XLM-based model i.e. CrossTCIN-xlm-roberta-msmarco as a cross-encoder and CrossTCIN-xlm-r-paraphrase as bi-encoder. Here, we use rank-based fusion where run 31, 32 and 33 uses RRF and run 34, 35 and 36 uses Borda fusion. For run 37, 38 and 39, no fusion method is used and we directly get the output from cross-encoder in the neural re-ranking stage.

- `gatenlp_en2es_run40` / `gatenlp_en2fr_run41` / `gatenlp_en2de_run42` / `gatenlp_en2es_run43` / `gatenlp_en2fr_run44` / `gatenlp_en2de_run45` / `gatenlp_en2es_run46` / `gatenlp_en2fr_run47` / `gatenlp_en2de_run48`: These are all bilingual runs where the language of query is English and the language of documents is depicted by ISO 639-1 code after `gatenlp_en2` identifier in the run name. Here we use multilingual models where the bi-encoder is CrossTCIN-xlm-r- paraphrase and cross-encoder is CrossTCIN-xlm-roberta-msmarco. For run 40, 41 and 42, RRF is used to calculate the final relevance score. Apart from this, all other runs use the final output from

cross-encoder but *key_conv* query is used in run 43, 44 and 45 and Udels query in run 46, 47 and 48.

- `gatenlp_en2es_run49` / `gatenlp_en2fr_run50` / `gatenlp_en2de_run51` / `gatenlp_en2es_run52` / `gatenlp_en2fr_run53` / `gatenlp_en2de_run54`: For these runs, we use cross-encoder as CrossTCIN-distilbert-multilingual-msmarco and bi-encoder as CrossTCIN-xlm-roberta-msmarco. Run 49, 50 and 51 uses Udels query and run 52, 53 and 54 uses *key_conv* as query for retrieving the documents.

**Other participant runs**. Besides our team, there are three more participants in the MLIA task 2, 1) Sinai (Universidad de Ja´en, Spain) 2) Cunimtir (Charles University, Czech Republic) 3) Ims (University of Padua, Italy) and there are in total 109 monolingual runs and 66 bilingual runs submitted. Sinai [36] specifically focused on Spanish language and used Lucene to do BM25 search using different fields of topics as a query on the index of different XML tag contents of the documents. Cunimtir's [37] monolingual runs employed language based Dirichlet model (run 1), per field normalisation weighting model (run 2) and two famous query expansion models i.e. Bose-Einstein model (run 3) and Kullback-Leibler divergence correct (run 4). For multilingual runs, they used neural machine translation models for translating the query into the document language before performing the retrieval. Ims [38] submitted multiple runs. Firstly, for each language, they submitted runs that use BM25 with default Lucene parameters where bm25 uses keyword field of the topic as query and c-bm25 use keyword and conversational formulation as a query. In addition to this, they also submitted run csum which is a one-stage CombSUM fusion of all the lexical runs, using only the keyword formulation of the query. In v-csum, they used a two-stage fusion to merge runs associated with query reformulations. For English runs, they submitted a few more additional runs such as nlex, a three-stage fusion using the topic formulations, lexical runs and neural runs, and nsle which uses a SLEDGE model [39] fine-tuned on the medical subset of the MSMARCO dataset to re-rank the documents.

## 6 Results and discussion

In this section, we explore the effectiveness of our approach. We evaluate the performance of Multistage BiCross Encoder using the relevance assessments provided by the MLIA organisers. All the submitted runs are evaluated using precision and normalised discounted cumulative gain (NDCG) that focus on top-ranked documents and, recall and R-precision (RPrec) whose focus is more on finding as many relevant documents as possible in all the retrieved documents.

### 6.1 Monolingual runs

Table 3 shows the results of the monolingual English runs. The first part of the table contains runs which retrieve 200 documents per query as these are a part of subtask 2 and the second part of the table contains runs which retrieve 1000 documents per query as this is a requirement for subtask 1. We focus on subtask 2 where we aim at achieving both high recall as well as high precision values for the least number of retrieved documents per topic query. Over and above, we include runs from both the subtasks so as to do a fair comparison of performance of our runs with all the submitted runs. As shown in Table 3, GATENLP runs outperform all other participant runs in almost all metrics by a significant margin (p-value<0.001 using paired t-test for all metrics) for subtask 2 runs. Amongst our runs, `gatenlp_run5` gives the highest scores, followed by `gatenlp_run3` and other runs shown in the table. The `gatenlp_run5` uses *key_conv* as a query and TCIN-stsb-roberta-large as a bi-encoder model.

**Table 3. Results for monolingual English runs.** Our runs have `gatenlp_` as a prefix in the name of the run. The first part of the table contains runs which retrieve 200 documents per query and the second part of the table contains runs which retrieve 1000 documents for each query. Best overall scores are highlighted in bold.

| Run ID | P@5 | P@10 | MAP | NDCG@10 | NDCG | Rprec | Recall |
|---|---|---|---|---|---|---|---|
| gatenlp_run5 | **0.9333** | **0.9000** | **0.2944** | **0.8331** | **0.5187** | **0.3486** | 0.4382 |
| gatenlp_run3 | 0.9200 | 0.8900 | 0.2912 | 0.8223 | 0.5155 | 0.3484 | 0.4375 |
| gatenlp_run2 | 0.9000 | 0.7967 | 0.2560 | 0.7775 | 0.4925 | 0.3215 | 0.4278 |
| gatenlp_run1 | 0.8867 | 0.8633 | 0.2776 | 0.8139 | 0.5067 | 0.3310 | **0.4411** |
| gatenlp_run7 | 0.8667 | 0.8800 | 0.2719 | 0.8212 | 0.5014 | 0.3305 | 0.4292 |
| CUNIMTIR_Run1 | 0.5933 | 0.4800 | 0.1145 | 0.4254 | 0.2802 | 0.1976 | 0.2613 |
| CUNIMTIR_Run3 | 0.3600 | 0.3233 | 0.0609 | 0.2712 | 0.1444 | 0.1046 | 0.1278 |
| CUNIMTIR_Run4 | 0.3533 | 0.3267 | 0.0530 | 0.2688 | 0.1422 | 0.0940 | 0.1239 |
| ims_bm25_1k | 0.3067 | 0.2433 | 0.0688 | 0.2391 | 0.2418 | 0.1579 | 0.2595 |
| ims_bm25_2k | 0.2400 | 0.1833 | 0.0478 | 0.1744 | 0.1789 | 0.1277 | 0.2028 |
| ims_bm25_3k | 0.2067 | 0.1633 | 0.0396 | 0.1413 | 0.1582 | 0.1075 | 0.1930 |
| ims_bm25_4k | 0.1933 | 0.1533 | 0.0367 | 0.1546 | 0.1483 | 0.1037 | 0.1677 |
| ims_nlex | **0.8933** | **0.9000** | **0.3055** | **0.8365** | 0.5740 | 0.3408 | 0.5593 |
| ims_c-bm25 | 0.8600 | 0.8267 | 0.2771 | 0.7592 | 0.5945 | 0.3089 | 0.6482 |
| ims_v-csum | 0.8533 | 0.8233 | 0.2999 | 0.7693 | **0.6092** | **0.3450** | **0.6516** |
| ims_bm25 | 0.7200 | 0.6900 | 0.2269 | 0.6202 | 0.5264 | 0.2673 | 0.6079 |
| CUNIMTIR_Run5. | 0.6867 | 0.6900 | 0.1908 | 0.5780 | 0.4574 | 0.2364 | 0.5160 |
| CUNIMTIR_Run1 | 0.6800 | 0.5033 | 0.1659 | 0.4928 | 0.4450 | 0.2223 | 0.5148 |
| ims_nsle | 0.5067 | 0.5133 | 0.1595 | 0.4084 | 0.4145 | 0.2205 | 0.4837 |
| CUNIMTIR_Run3 | 0.4800 | 0.3367 | 0.0882 | 0.2944 | 0.2500 | 0.1221 | 0.2986 |
| CUNIMTIR_Run2 | 0.4667 | 0.3033 | 0.0646 | 0.3005 | 0.2379 | 0.1163 | 0.2683 |
| CUNIMTIR_Run4 | 0.4267 | 0.3400 | 0.0658 | 0.2809 | 0.2200 | 0.1051 | 0.2662 |

This suggests that the use of bi-encoder model pre-fine-tuned on STS data proved to be beneficial when compared to the ones pre-fine-tuned on MSMARCO dataset for monolingual English runs. Regarding the type of query, the results show that employing *key_conv* as a query achieves large gains as compared to Udels query and *t5_query* method. Furthermore, *t5_query* got comparatively lower results which suggests that doing retrieval using concatenated reworded queries induces noise in retrieved documents. Even though subtask 1 runs retrieve 1000 documents per query, our runs still perform equally well and in some cases even surpass subtask 1 runs despite the fact that we only retrieve 200 documents for each query. This indicates that retrieving more documents leads to high recall but there is minimal difference in performance for precision focused metrics.

For monolingual Spanish (Table 4), our runs outperformed all other submissions, to a statistically significant degree (paired t-test p-value<0.001 for all metrics). `gatenlp_run37` scores highest in precision whereas `gatenlp_run25` gave the best results for NDCG and recall. Similarly, Tables 5 and 6 shows the results for monolingual French and monolingual German respectively. Although ours is the only team that submitted monolingual French runs, we achieve highly competent scores. Overall, we find that the runs which use weighted Comb-SUM on model CrossTCIN-xlm-r-paraphrase (bi-encoder) and CrossTCIN-distilbert-multi-lingual-msmarco (cross-encoder) give top scores for recall and NDCG. Additionally, we also speculate the performance of multilingual models on different languages and we found that for most of the metrics, the scores of German runs are comparatively higher, followed by French and Spanish runs respectively. These differences might arise from intrinsic language variations in the pre-training of multilingual transformer models (eg. mBERT, XLM-RoBERTa

**Table 4. Results for monolingual Spanish runs.** Our runs have `gatenlp_` as a prefix in the name of the run. The first part of the table contains runs which retrieve 200 documents per query and the second part of the table contains runs which retrieve 1000 documents for each query. Best overall scores are highlighted in bold.

| Run ID | P@5 | P@10 | MAP | NDCG@10 | NDCG | Rprec | Recall |
|---|---|---|---|---|---|---|---|
| gatenlp_run37 | **0.8333** | **0.7933** | 0.2043 | 0.7263 | 0.3705 | 0.2806 | 0.3086 |
| gatenlp_run25 | 0.8133 | 0.7767 | 0.2154 | 0.7455 | **0.3808** | 0.2795 | **0.3111** |
| gatenlp_run28 | 0.8067 | 0.7767 | 0.2113 | **0.7478** | 0.3768 | 0.2758 | 0.3086 |
| gatenlp_run34 | 0.7933 | 0.7833 | 0.2173 | 0.7383 | 0.3769 | 0.2858 | 0.3086 |
| gatenlp_run31 | 0.7933 | 0.7867 | **0.2246** | 0.7362 | 0.3790 | **0.2873** | 0.3086 |
| sinai_sinai1 | 0.5200 | 0.4867 | 0.0900 | 0.4629 | 0.2177 | 0.1557 | 0.1767 |
| sinai_sinai2 | 0.4400 | 0.4067 | 0.0631 | 0.3868 | 0.1835 | 0.1284 | 0.1537 |
| sinai_sinai4 | 0.3600 | 0.3067 | 0.0535 | 0.2904 | 0.1738 | 0.1243 | 0.1594 |
| sinai_sinai3 | 0.2267 | 0.1733 | 0.0284 | 0.1820 | 0.1121 | 0.0786 | 0.1011 |
| sinai_sinai5 | 0.2267 | 0.1733 | 0.0155 | 0.1832 | 0.0634 | 0.0407 | 0.0444 |
| ims_bm25_1k | 0.2067 | 0.1867 | 0.0577 | 0.1812 | 0.1944 | 0.1366 | 0.2142 |
| ims_bm25_2k | 0.2000 | 0.1800 | 0.0591 | 0.1745 | 0.2003 | 0.1402 | 0.2275 |
| ims_bm25_3k | 0.1733 | 0.1433 | 0.0444 | 0.1359 | 0.1744 | 0.1196 | 0.2072 |
| ims_bm25_4k | 0.0867 | 0.0800 | 0.0309 | 0.0793 | 0.1535 | 0.1046 | 0.1900 |
| ims_c-bm25 | **0.7000** | 0.6933 | 0.1654 | 0.6346 | **0.3993** | 0.2224 | **0.4084** |
| ims_v-csum | 0.6867 | **0.7133** | 0.1697 | **0.6604** | 0.3797 | 0.2171 | 0.3612 |
| ims_csum | 0.6800 | 0.6200 | **0.1720** | 0.5822 | 0.3769 | **0.2259** | 0.3779 |
| ims_bm25 | 0.6133 | 0.5800 | 0.1458 | 0.5263 | 0.3540 | 0.2020 | 0.3740 |
| sinai_sinai1 | 0.5200 | 0.4867 | 0.1000 | 0.4629 | 0.2839 | 0.1560 | 0.2928 |
| sinai_sinai2 | 0.4400 | 0.4067 | 0.0715 | 0.3868 | 0.2436 | 0.1285 | 0.2618 |
| sinai_sinai4 | 0.3600 | 0.3067 | 0.0626 | 0.2904 | 0.2368 | 0.1247 | 0.2689 |
| sinai_sinai5 | 0.2267 | 0.1733 | 0.0157 | 0.1832 | 0.0693 | 0.0408 | 0.0550 |
| sinai_sinai3 | 0.2267 | 0.1733 | 0.0342 | 0.1820 | 0.1644 | 0.0788 | 0.1906 |

etc) or due to differences in the document processing pipeline of different languages or by both of these factors. At last, the results show that neural methods perform far better than the lexical-based BM25 baselines such as the ones used in `ims_c-bm25` and `ims_bm25`.

## 6.2 Bilingual runs

Table 7 compares the performance of our different bilingual runs. These include English to Spanish (en2es), English to French (en2fr) and English to German (en2de) runs. The best performing run in each case attains P@5≥0.8, depicting that there is at least an average of 80% chance of getting a relevant document in the top 5 retrieved documents even when the language of query and document is different. As all runs retrieve a total of 200 documents for each query, the recall value remains similar for all three bilingual cases, which shows that the

**Table 5. Results for monolingual French runs.** All runs retrieve 200 documents per query as there were no runs submitted by any team which retrieve 1000 documents per query. Best overall scores are highlighted in bold.

| Run ID | P@5 | P@10 | MAP | NDCG@10 | NDCG | Rprec | Recall |
|---|---|---|---|---|---|---|---|
| gatenlp_run26 | **0.8800** | **0.7533** | **0.3505** | **0.7490** | **0.5672** | **0.3773** | **0.5267** |
| gatenlp_run29 | 0.8600 | 0.7400 | 0.3302 | 0.7324 | 0.5406 | 0.3651 | 0.4926 |
| gatenlp_run32 | 0.8133 | 0.7367 | 0.3161 | 0.7180 | 0.5297 | 0.3593 | 0.4926 |
| gatenlp_run35 | 0.8133 | 0.7267 | 0.3125 | 0.7116 | 0.5268 | 0.3541 | 0.4926 |
| gatenlp_run38 | 0.7867 | 0.6400 | 0.2752 | 0.6436 | 0.5030 | 0.3269 | 0.4926 |

**Table 6. Results for monolingual German runs.** Our runs have `gatenlp_` as a prefix in the name of the run. The first part of the table contains runs which retrieve 200 documents per query and the second part of the table contains runs which retrieve 1000 documents for each query. Best overall scores are highlighted in bold.

| Run ID | P@5 | P@10 | MAP | NDCG@10 | NDCG | Rprec | Recall |
|---|---|---|---|---|---|---|---|
| gatenlp_run30 | **0.9067** | **0.8767** | 0.4537 | **0.8234** | 0.6403 | 0.4794 | 0.6253 |
| gatenlp_run27 | 0.9000 | 0.8667 | **0.4629** | 0.8211 | **0.6488** | 0.4858 | **0.6339** |
| gatenlp_run36 | 0.8733 | 0.8267 | 0.4442 | 0.7772 | 0.6377 | 0.4843 | 0.6253 |
| gatenlp_run33 | 0.8733 | 0.8300 | 0.4531 | 0.7793 | 0.6399 | **0.4972** | 0.6253 |
| gatenlp_run39 | 0.7733 | 0.7700 | 0.4227 | 0.7078 | 0.6200 | 0.4601 | 0.6253 |
| ims_bm25_1k | 0.1667 | 0.1633 | 0.0700 | 0.1475 | 0.2288 | 0.1413 | 0.3063 |
| ims_bm25_2k | 0.1667 | 0.1600 | 0.0793 | 0.1515 | 0.2176 | 0.1388 | 0.2769 |
| ims_bm25_4k | 0.1467 | 0.1433 | 0.0629 | 0.1396 | 0.1967 | 0.1120 | 0.2589 |
| ims_bm25_3k | 0.1400 | 0.1367 | 0.0650 | 0.1276 | 0.1924 | 0.1163 | 0.2488 |
| ims_v-csum | **0.7267** | **0.6733** | **0.3447** | **0.6341** | **0.6174** | **0.3737** | 0.7080 |
| ims_csum | 0.6267 | 0.5700 | 0.3072 | 0.5315 | 0.5731 | 0.3507 | 0.6940 |
| ims_c-bm25 | 0.6133 | 0.5633 | 0.2890 | 0.5150 | 0.5667 | 0.3131 | **0.7114** |
| ims_bm25 | 0.5933 | 0.5333 | 0.2869 | 0.4912 | 0.5572 | 0.3173 | 0.6924 |

MLIA corpus does not contain many relevant documents in Spanish, French, and German language. If we compare the performance of run 43, 44 and 45 with run 46, 47 and 48, we see that the former runs, which use *key_conv* as query, give better results than the latter ones which use Udels query. We see similar results for run 49, 50 and 51 which use Udels query and run 52, 53 and 54 which use *key_conv* as a query. This shows that the use of keyword and conversational formulation as a query gives the best results for bilingual runs (p-value≤0.05 for paired t-test). For English to German runs, `gatenlp-en2de-run42` has the top scores for all the metrics. Apart from this, we couldn't find any single model which performs well for all the languages as different methods give distinct results for different metrics and there is considerable variability as shown in Table 7. Although ours were the only runs submitted for the above given bilingual pairs, the evaluation results show that using BiCross encoder by machine

**Table 7. Results for bilingual Spanish (es), French (fr) and German (de) runs.** Here the language of query is English and language of documents is depicted by ISO 639-1 code after `gatenlp_en2` identifier in the run name. All runs retrieve 200 documents per query. Best overall scores are highlighted in bold.

| Run ID | P@5 | P@10 | MAP | NDCG@10 | NDCG | Rprec | Recall |
|---|---|---|---|---|---|---|---|
| gatenlp_en2es_run49 | **0.8533** | 0.7367 | 0.1579 | 0.7042 | 0.3214 | 0.2273 | **0.2565** |
| gatenlp_en2es_run52 | 0.8200 | 0.7700 | 0.1666 | **0.7368** | **0.3287** | 0.2286 | **0.2565** |
| gatenlp_en2es_run40 | 0.8067 | **0.7733** | **0.1740** | 0.7211 | 0.3277 | **0.2380** | **0.2565** |
| gatenlp_en2es_run43 | 0.8000 | 0.6867 | 0.1538 | 0.6555 | 0.3155 | 0.2287 | **0.2565** |
| gatenlp_en2es_run46 | 0.7733 | 0.6367 | 0.1439 | 0.6330 | 0.3120 | 0.2231 | **0.2565** |
| gatenlp_en2fr_run53 | **0.8400** | **0.7467** | **0.2870** | **0.7245** | **0.4993** | 0.3220 | **0.4452** |
| gatenlp_en2fr_run41 | 0.8267 | 0.7033 | 0.2811 | 0.6980 | 0.4966 | **0.3294** | **0.4452** |
| gatenlp_en2fr_run44 | 0.7667 | 0.6667 | 0.2527 | 0.6622 | 0.4801 | 0.3107 | **0.4452** |
| gatenlp_en2fr_run47 | 0.7400 | 0.6300 | 0.2378 | 0.6234 | 0.4633 | 0.2980 | 0.4360 |
| gatenlp_en2fr_run50 | 0.7133 | 0.6700 | 0.2506 | 0.6521 | 0.4712 | 0.3054 | 0.4360 |
| gatenlp_en2de_run42 | **0.8000** | **0.7600** | **0.2776** | **0.7300** | **0.4546** | **0.3307** | **0.3950** |
| gatenlp_en2de_run54 | 0.7733 | 0.7267 | 0.2680 | 0.7007 | 0.4484 | 0.3221 | **0.3950** |
| gatenlp_en2de_run51 | 0.7200 | 0.6867 | 0.2475 | 0.6568 | 0.4334 | 0.3029 | 0.3907 |
| gatenlp_en2de_run45 | 0.7133 | 0.6700 | 0.2474 | 0.6444 | 0.4349 | 0.3076 | **0.3950** |
| gatenlp_en2de_run48 | 0.6867 | 0.6433 | 0.2292 | 0.6099 | 0.4187 | 0.2978 | 0.3907 |

translating query into document language helped in attaining competitive baselines for future research.

On the whole, domain-specific fine-tuning of transformer models on TC+IFCN dataset gave a boost in performance. Also, the results suggest that fine-tuning multilingual models such as mBERT and XLM-RoBERTa on Cross_TC+IFCN can make models quickly adapt to the domain-specific data and transfer relevance matching across languages. This is coherent with the previous work [40]. In spite of the fact that our training dataset consists mainly of scientific scholarly research papers, our fine-tuned models were able to transfer knowledge to articles about general COVID-19 health-related content and thereby accomplishing promising results on the MLIA corpus. It is also worth emphasising the effectiveness of BiCross encoder to refine and re-rank the candidate documents which has the dual advantage of high precision and high recall values in both monolingual and bilingual runs. Regarding the computational complexity of BiCross encoder, it depends on the number of documents to be re-ranked by the cross-encoder in the neural re-ranking stage and this can be controlled by restricting the count of top $n$ documents retrieved from neural refinement stage. The benefit of the neural refinement stage is that the representation from the bi-encoder can be stored locally which obviates the need for an inference pass to the encoder model during re-ranking and this process can be expedited with GPU based implementation of similarity search such as FAISS [14].

## 7 Conclusion

This paper proposed a novel Multistage BiCross Encoder developed for the MLIA COVID-19 multilingual semantic search (task 2). As detailed above, the multistage BiCross encoder is a three-stage approach consisting of an initial retrieval using Okapi BM25 algorithm followed by a transformer-based bi-encoder and cross-encoder to effectively rank the documents using sentence-level score aggregation with respect to the query. Our method exploited transfer learning, by fine-tuning large pre-trained transformer models on domain-specific data for retrieving COVID-19 health-related articles in multiple languages. While the approach is conceptually simple, the independently evaluated MLIA results demonstrate that the use of bi-encoder and cross-encoder along with BM25 is highly effective in outperforming other state-of-the-art methods according to a wide range of metrics, and it has the twofold benefit of high precision in retrieving the top-ranked documents (P@5≥0.8 for best performing run in each case), as well as a high recall for all retrieved documents. We also find that employing keyword and conversational formulation as a query gives the highest scores in both monolingual and bilingual search settings. We hope that our research will help improve multilingual access to reliable COVID-19 health information thereby mitigating the impact of the 'infodemic' as a consequece of the ongoing COVID-19 pandemic.

Future work will experiment with further hyperparameter tuning and making additional improvements to the neural architecture. We also plan to test the Multistage BiCross Encoder on other document retrieval and similar information access tasks.

## Author Contributions

**Conceptualization:** Iknoor Singh.

**Formal analysis:** Iknoor Singh.

**Funding acquisition:** Kalina Bontcheva.

**Methodology:** Iknoor Singh.

**Project administration:** Carolina Scarton, Kalina Bontcheva.

**Resources:** Iknoor Singh.

**Supervision:** Carolina Scarton, Kalina Bontcheva.

**Validation:** Iknoor Singh.

**Visualization:** Iknoor Singh.

**Writing – original draft:** Iknoor Singh.

**Writing – review & editing:** Carolina Scarton, Kalina Bontcheva.

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
