## [Decision Letter · Decision Letter 0]

16 Jun 2021

PONE-D-21-17874

Multistage BiCross Encoder for Multilingual Access to COVID-19 Health Information

PLOS ONE

Dear Dr. SINGH,

Thank you for submitting your manuscript to PLOS ONE. After careful consideration, we feel that it has merit but does not fully meet PLOS ONE’s publication criteria as it currently stands. Therefore, we invite you to submit a revised version of the manuscript that addresses the points raised during the review process.

Based on the comments received from the reviewers and my own observation, I recommend minor revisions for the article.

We look forward to receiving your revised manuscript.

Kind regards,

Thippa Reddy Gadekallu

Academic Editor

PLOS ONE

Journal Requirements:

Reviewers' comments:

Reviewer's Responses to Questions

**Comments to the Author**

1. Is the manuscript technically sound, and do the data support the conclusions?

Reviewer #1: Partly

Reviewer #2: Yes

2. Has the statistical analysis been performed appropriately and rigorously? 

Reviewer #1: Yes

Reviewer #2: Yes

3. Have the authors made all data underlying the findings in their manuscript fully available?

Reviewer #1: Yes

Reviewer #2: Yes

4. Is the manuscript presented in an intelligible fashion and written in standard English?

Reviewer #1: Yes

Reviewer #2: Yes

5. Review Comments to the Author

Reviewer #1: Authors should add a list of contribtion before the organization paragraph in introduction.

General information that is well known should be reduced.

- All the key terms of the equations must be mentioned

- There must be column names in the tables with all borders.

- There are many typos and grammatical mistakes in the entire paper.

- The authors should further add explanation about research method.

- What are the computational resources reported in the state of the art for the same purpose?

- Major contribution was not clearly mentioned in the conclusion part.

- The discussion is very important in research paper. Nevertheless, this section is short and should be presented completely.

- Authors should add the most recent reference:

1) Classification of COVID-19 individuals using adaptive neuro-fuzzy inference system, Multimedia Systems, 1-15

2) Classification and Categorization of COVID-19 Outbreak in Pakistan, CMC

Reviewer #2: Introduction needs to explain the main contributions of the work more clearly.

The authors should emphasize the difference between other methods to clarify the position of this work further.

The authors should define all the notations and acronyms before using them.

In Abstract the authors can mention there achieved results.

The authors can refer the latest works on covid. Deep learning and medical image processing for coronavirus (COVID-19) pandemic: A survey

Multistage BiCross Encode need more explanation.

6. PLOS authors have the option to publish the peer review history of their article (what does this mean?). If published, this will include your full peer review and any attached files.

Reviewer #1: No

Reviewer #2: No

---

## [Author Response · Author response to Decision Letter 0]

6 Jul 2021

Dear Editor PLOS One,

Thank you for giving us the opportunity to submit a revised draft of our manuscript entitled “Multistage BiCross encoder for multilingual access to COVID-19 health information” to PLOS One. We appreciate the time and effort that you and the reviewers have dedicated to give valuable feedback on our manuscript. We have been able to incorporate changes to reflect the suggestions provided by the reviewers. We have highlighted all the changes within the manuscript. 

Here is a point-by-point response to the reviewers’ comments and concerns. 

Comments from Reviewer 1:

 - Authors should add a list of contribution before the organization paragraph in introduction. 

We have added a list of contributions in the introduction section (Page 2)

- General information that is well known should be reduced.

We rewrote parts of the paper, removing all the unnecessary information

- All the key terms of the equations must be mentioned 

We have revised all the equations, aiming to make them more comprehensible. All the changes can be seen in the highlighted part of the manuscript after the equations (Page 6-9).

- There must be column names in the tables with all borders.

We have added the borders to all the tables.

- There are many typos and grammatical mistakes in the entire paper.

We have revised the complete manuscript multiple times and have corrected all typos and grammatical mistakes.

- The authors should further add explanation about research method.

We have added more explanation and details, aiming to make the methodology more clear. We have also added some more details about our method in the introduction section. (Page 2)

- What are the computational resources reported in the state of the art for the same purpose?

You have raised an important point here. We have added details in the last paragraph of the “Results and Discussion” section and have also highlighted it (Page 15). However, in the case of our study, detailed comparison seems slightly out of scope because there are currently no reported computational resources by other state-of-the-art on the same MLIA dataset. Nevertheless, this is an interesting research direction and we hope to explore this more in our future work.

- Major contribution was not clearly mentioned in the conclusion part.

 We have revised the conclusion section and have tried to incorporate all major contributions. (Page 15)

- The discussion is very important in a research paper. Nevertheless, this section is short and should be presented completely.

We have added more details to the discussion section. All the major changes are highlighted in the “Results and discussion” part. (Page 11-15)

- Authors should add the most recent reference:1) Classification of COVID-19 individuals using adaptive neuro-fuzzy inference system, Multimedia Systems, 1-15 2) Classification and Categorization of COVID-19 Outbreak in Pakistan, CMC

Thanks for letting me know about these references. We have cited it in our manuscript. (Page 5)

Comments from Reviewer 2:

 - Introduction needs to explain the main contributions of the work more clearly. 

We have added a list of contributions in the introduction section. We have also added some more details about our method in the introduction section. All changes have been highlighted (Page 2).

 - The authors should emphasize the difference between other methods to clarify the position of this work further.

We have clarified the differences between our work and other methods in the “Related work” section. These have been highlighted (Page 4-5).

 - The authors should define all the notations and acronyms before using them.

We revised our manuscript to incorporate the required changes (highlighted in the manuscript). 

 - In Abstract the authors can mention there achieved results.

We added this information to the abstract. 

 - The authors can refer the latest works on covid. Deep learning and medical image processing for coronavirus (COVID-19) pandemic: A survey

We have cited it in our manuscript (Page 5). Thanks.

 - Multistage BiCross Encode need more explanation.

The detailed description of our method has been given in section 4 of the paper (Page 5-9). We have also added some more details in various sections of the paper to make it more comprehensible (these have been highlighted). 

In addition to the above comments, all spelling and grammatical errors have been corrected. We look forward to hearing from you in due time regarding our submission and to respond to any further questions and comments you may have. 

Thank you and best regards.

Yours Sincerely,

Iknoor Singh

Department of Computer Science

The University of Sheffield, UK

E-mail: i.singh@sheffield.ac.uk

---

## [Decision Letter · Decision Letter 1]

18 Aug 2021

Multistage BiCross encoder for multilingual access to COVID-19 health information

PONE-D-21-17874R1

Dear Dr. SINGH,

We’re pleased to inform you that your manuscript has been judged scientifically suitable for publication and will be formally accepted for publication once it meets all outstanding technical requirements.

Kind regards,

Balaraman Ravindran, Ph.D.

Academic Editor

PLOS ONE

Additional Editor Comments (optional):

Reviewers' comments:

Reviewer's Responses to Questions

**Comments to the Author**

1. If the authors have adequately addressed your comments raised in a previous round of review and you feel that this manuscript is now acceptable for publication, you may indicate that here to bypass the “Comments to the Author” section, enter your conflict of interest statement in the “Confidential to Editor” section, and submit your "Accept" recommendation.

Reviewer #1: (No Response)

2. Is the manuscript technically sound, and do the data support the conclusions?

Reviewer #1: Yes

3. Has the statistical analysis been performed appropriately and rigorously? 

Reviewer #1: Yes

4. Have the authors made all data underlying the findings in their manuscript fully available?

Reviewer #1: Yes

5. Is the manuscript presented in an intelligible fashion and written in standard English?

Reviewer #1: Yes

6. Review Comments to the Author

Reviewer #1: The authors have addressed almost all my suggestions. I would like to accept this paper.

7. PLOS authors have the option to publish the peer review history of their article (what does this mean?). If published, this will include your full peer review and any attached files.

Reviewer #1: No

---

## [Editor Report · Acceptance letter]

27 Aug 2021

PONE-D-21-17874R1 

Multistage BiCross encoder for multilingual access to COVID-19 health information 

Dear Dr. Singh:

I'm pleased to inform you that your manuscript has been deemed suitable for publication in PLOS ONE. Congratulations! Your manuscript is now with our production department. 

Kind regards, 

on behalf of

Dr. Balaraman Ravindran 

Academic Editor

PLOS ONE